# Knowledge and practice of clients on preventive measures of COVID-19 pandemic among governmental health facilities in South Wollo, Ethiopia: A facility-based cross-sectional study

**Reta Dewau** [1], **Tefera Chane Mekonnen** [2], **Sisay Eshete Tadesse** [2], **Amare Muche** [1], **Getahun Gebre Bogale** [3], **Erkihun Tadesse Amsalu** [1] *

**1** Department of Epidemiology and Biostatistics, school of public health, College of Medicine and Health sciences, Wollo University, Dessie, Ethiopia, **2** Department of nutrition and dietetics, school of public health, College of Medicine and Health sciences, Wollo University, Dessie, Ethiopia, **3** Department of Health informatics, school of public health, College of Medicine and Health sciences, Wollo University, Dessie, Ethiopia

* brhaneyared07@gmail.com

## Abstract

### Introduction

Coronavirus-19 is a global health challenge and need an immediate action. Thus, understanding client's knowledge about SARS-COV2 causes, roots of transmissions, and prevention strategies are urgently warranted. Although there were global studies reported knowledge and preventive practices of COVID-19, but the information is not representative and inclusive for Ethiopia. Thus, the current study is done to identify the knowledge and the prevention strategies for COVID-19 among clients in South Wollo, Ethiopia.

### Methods

An institutional based cross-sectional study was conducted from May 21 to 30, 2020 among clients seeking service in Dessie town health facilities. A total of 81 clients were included from the selected health facilities with simple random sampling technique. We developed measuring tools by adopting from World Health Organization and center for disease prevention recommendation manual for assessing service providers' knowledge and preventive practices. For data entry Epi-data 3.1 version was employed and further data management and analysis was performed using STATA Version 14. Student T-test and one way ANOVA were computed to see the mean difference in knowledge and practice between and among the group. Chi-square test was also done to portray the presence of association between different co-variants with client's knowledge and preventive practices.

### Results

Findings of the study showed that more than half (56.8%) of the participants had good knowledge about its symptoms, way of spread and prevention of the virus. Furthermore,

**Data Availability Statement:** All relevant data are within the paper and its Supporting information files.

**Funding:** The author(s) received no specific funding for this work.

**Competing interests:** The authors have declared that no competing interests exist.

**Abbreviations:** CDC, Center for Diseases Prevention and Control; COVID-19, Coronavirus Disease 19; DM, Diabetes Mellitus; HIV, Human Immune Deficiency Virus; IQR, Inter-Quartile Range; KAP, Knowledge, attitude and Practice; PPE, Personal Protective Equipment's; SD, Standard Deviation; WHO, World Health Organization.

65.4% of clients demonstrated five or more preventive practice measures of COVID-19. The mean preventive practice score with standard deviation was (4.75±1.28 from 6 components). In the current study, knowledge had no significant difference among sex, education status, and monthly income. However, COVID-19 transmission knowledge was significantly higher among urban residents. Thus, clients who were knowledgeable about way of transmission and symptoms of COVID-19 had significantly higher COVID-19 preventive practice.

## Conclusion

Our findings revealed that clients' knowledge and preventive practice of COVID-19 were not optimal. Clients with good knowledge and urban residents had practiced better prevention measures of the pandemic, signifying that packages and programs directed in enhancing knowledge about the virus is useful in combating the pandemic and continuing safe practices.

## Introduction

The COVID-19 is caused by SARS CoV-2 virus which belongs to the genus of Beta-corona virus [1]. Studies showed that the structural similarity with SARS-CoV [2] but it is severe than SARS-CoV [3]. By 20th of June 2020, 8,567,280 total cases and 455,549 deaths have been reported globally [4, 5]. According to reports the basic reproduction number (R0) of the virus has been varied which may range from 2.2 to 4.82 [6].

It is a respiratory illness and primarily transmitted by person to person contact and droplets from the respiratory tract. According to recent studies there is asymptomatic transmission of the virus [7]. Thus, for high risk individuals especially for those with co-morbidities vaccination is strongly advised [8].

Regarding symptoms, infected individuals manifest symptoms vary from pauci-symptomatic forms to symptomatic signs of respiratory insufficiency due to this infected individuals require mechanical ventilation in an intensive care unit (ICU). Affected individuals also manifest systemic signs including septic shock, sepsis, and organ failure [9].

The victims of COVID-19 in resource limited setting mainly Africa; the number of health care workers will significantly increase and have devastating impact. As of 4th March 2020 the first case detected in Ethiopia, the total cases reached to 3954 and 65 deaths by 19 June 2020 and the pandemic has been exponentially spreading. Amhara region is one of among the high hot spot areas in the nation and shared about 252 cases and 2 deaths, of which South Wollo Zone responsible for 12 case of the COVID-19 regional burden [10].

Adherences to control the pandemic according to the guiding protocol are essential, which are hugely influenced by their level of understanding and mitigation strategies of the virus [11]. Despite of the overwhelming impact of the pandemic, COVID-19, updated scientific evidences on clients seeking service in health institutions have not been documented and assessed in Ethiopia, including our study area; South Wollo. Hence, a clear understanding about pathophysiology, transmission modality, sign and/ symptoms, and prevention and control measures are the basis and core spectrums to tackle the pandemic as outlined in the research road map of World Health Organization. Therefore, the current research aims to explore client's level of understanding and preventive strategies of the pandemic in the selected health facilities of South Wollo, North east Ethiopia.

## Methodology

### Study setting and design

A facility based cross-sectional study was conducted in the selected health facilities of Dessie town for assessing the knowledge and preventive practices of clients to manage the current pandemic, COVID-19 from May 21 to May 30, 2020. The town is 401 KM (kilometer) far from Addis Ababa the capital city of Ethiopia. According to 2007 national census, about 151,174 populations found in the city administration of which 72,932 are men and 78,242 women. The area is found at an elevation of 2,470 meters (8,100 feet) above sea level. Regarding public health facilities there are 1 Referral hospital, 1 general hospital, and 6 health centers with a catchment population of around 10 million.

### Populations and sample

For this study, clients seeking service in the selected health facilities found in Dessie town was the target population and those clients seeking service in the selected health facilities during data collection period was considered as the study population. Study subjects who are able to communicate and seeking service in the selected health facilities were eligible for this study under investigation.

We calculated the sample size using score test for one sample proportion by STATA version 14 with the assumption that 50% of clients had good knowledge that scored above the mean, 5% level of significance ($\alpha$), power of 80% ($\beta$), and at least 16% of client's knowledge improvement is expected after six months of the diseases duration (December to May). The total calculated sample size was 83 clients.

We selected Dessie town purposively from south wollo zone since it is hotspot for the spread of the virus as speculated by the Ethiopian Public Health Research Institute and selected by the federal ministry of health of Ethiopia as the only treatment and screening center for COVID-19 during the early phase of the pandemic in the country. We selected 3 health facilities namely Dessie Referral hospital, Segno Gebeya health center, and Buanbuawuha health centers in Dessie town randomly using lottery method. Then, we proportionally allocated the participants to each health facility based on the patient flow of the facilities. Then we selected 43 participants from Dessie referral hospital, 20 from Segnogebeya health center, and 20 from Buanbuawuha health center consecutively.

### Data measurement

A semi-structured questionnaire developed from World Health Organization (WHO) protocol was used to collect the data via face-to-face. About 4 data collectors and 3 supervisors to handle the field work were recruited after providing intensive training.

The data comprised of socio-demographic features of participants, Knowledge and practice of clients about COVID-19. The outcomes variables of the study were knowledge and preventive practice. We used items included from CDC and WHO hospitals monitoring protocol to measure knowledge and practices computed by factor scores.

Knowledge in this study shows the knowledge of clients towards COVID-19 pandemic. Accordingly, those clients who responded above the mean score value from knowledge related questions (18 questions) was considered as having good knowledge towards COVID-19. And those participants who responded below the mean score value from knowledge related questions were considered as having poor knowledge towards COVID19.

Preventive Practice in this study shows the practice of clients towards prevention of COVID-19 pandemic. Accordingly, those clients who responded above the mean score value

from practice related questions (6 questions) were considered as having good practice and those participants who answered below the mean score value from practice related questions were considered as having poor practice towards COVID-19 prevention.

## Statistical analysis and data management

For data entry we used Epi-data 3.1 and STATA Version 14 used in performing the statistical analysis. Summary statistics was computed to express the condition of the pandemic in terms of socio-demographic, knowledge, and preventive practices. Internal comparison was also made to see variability among health facility. Results were presented using text, table and figures. To see the mean difference in Knowledge and practice between and among the group we computed T-test and one way ANOVA. Additionally, to portray the association between co-variants with client's knowledge and preventive strategies a Chi-square test of association was used.

## Ethics approval and consent to participate

Ethical clearance was obtained from the ethical review committee of the College of Medicine and Health Sciences of Wollo University. All study participants were reassured that name was not needed and would not be recorded. Chances were given to ask anything about the study and being made free to refuse or stop the interview at any moment they want. Written informed consents were obtained from the study participants. Confidentially of the participants was ensured throughout the study period.

# Results

## Socio-demographic and medical profiles of the clients

In this study, from a total of 83 participants about 81 clients attending governmental health facilities were included in the analysis with a response rate of 97.6%. More than half of the participants, about 44 (54.32%), of them were females. Regarding residency more than three-fourths, 62 (76.54) of the clients were urban dwellers. Majority of the clients heard about COVID-19 from broad cast about the virus. The median age of the clients was 32 years old (IQR = 14). The median monthly income was 3000 Ethiopian birr (ETB) with IQR = 2000. More than three-fourths, 66 (77.78%) of respondents were married. About 26 (32.10%) of the participants were merchants followed by government employees, 24 (29%) (Table 1).

From the total study participants, about 71 (87.65%), 6(8.6%) and 3(3.7%) of clients got information about COVID-19 for the first time from broadcast medias of the country, health care workers and social media respectively.

## Knowledge of clients about COVID-19

From the total study participants, 67 (82.72%) heard about the novel coronavirus (COVID-19). About 47 (69%) of them had knowledge about the cause of COVID-19 infection. Nearly all 79 (97.5%) of the respondents understood that COVID-19 is a contagious disease. Majority of them 79 (88.89%) were responded that COVID-19 has not yet any treatment/vaccine. Similarly the majority of the clients, about 77 (95%) knew PPEs (personal protective equipment's) used for COVID-19. The risk conditions of COVID-19 was known by 63(67.78%) of them. About 35 (43%), of the participants complained stress/anxiety due to COVID-19 (Table 2).

Over all the majority of clients, about 46(56.79%) of them had good knowledge about COVID-19 and about 35(43.21%) of them had poor knowledge about the Disease in the study area.

**Table 1. Socio-demographic characteristics of clients in Dessie town public health facilities, Ethiopia, 2020 (n = 81).**

| Variables | Category | Frequency (%) | Knowledgescore (mean±standard deviation) from 18 | t/F statistic | p-value |
|---|---|---|---|---|---|
| Sex | Male | 37 (45.68) | 14.41±1.76 | 0.10 | 0.94 |
| | Female | 44 (54.32) | 14.36±1.94 | | |
| Educational status | No formal education | 22 (27.16) | 14.59±1.30 | 1.23 | 0.31 |
| | Primary education | 27 (33.33) | 13.89±2.42 | | |
| | Secondary education | 16 (19.75) | 14.94±1.24 | | |
| | Diploma and above | 16 (19.75) | 14.38±1.82 | | |
| Residence | Urban | 62 (76.54) | 14.53±1.94 | 1.32 | 0.19 |
| | Rural | 19 (23.46) | 13.89±1.45 | | |
| Marital status | Married | 63 (77.78) | 14.26±1.72 | -1.03 | 0.30 |
| | Currently Single | 18(22.22) | 14.78±2.26 | | |
| Occupation | Farmer | 15 (18.52) | 14.07±1.34 | 0.97 | 0.41 |
| | Merchant | 26 (32.10) | 14.89±1.40 | | |
| | Daily labor | 16 (19.75) | 14.25±2.79 | | |
| | Government employ | 24 (29.63) | 14.12±1.78 | | |
| Infant/children/elders at home | Yes | 35 (43.21) | 14.11±1.83 | -1.14 | 0.26 |
| | No | 46 (56.79) | 14.59±1.86 | | |

Majority of the study subjects know about ways of COVID-19 transmission; sneezing 76 (93.8%), coughing 78 (96%), contact 78 (96%), and shaking 65 (80%). Still 24 (30%), and 17 (21%) of the clients had no knowledge on the role of self-isolation and face mask in preventing the virus person to person transmission respectively. About 16 (20%) of the respondents had no information whether hand shaking can transmit the disease.

In addition, the clients feel, perceive or develop symptoms of distress or anxiety like headache, palpitation, and worry of too much and spell of terror or panic. They alleviate the stress/anxiety feelings with different activities like praying, doing sport, listening music, reading books, discussion with families and calling to health care providers.

## Practice of clients towards COVID-19 prevention

Preventive practices of the respondents against COVID-19 have statistical significant association with overall respondents good Knowledge score; mask utilizations (Pearson $chi^2$ (1) = 13.43, 0.001), avoid social distancing (Pearson $chi^2$ (1) = 9.39, 0.002), isolation of cases (Pearson $chi^2$ (1) = 13.43, 0.001) and overall preventive practice (Pearson $chi^2$ (1) = 26.43, 0.001) (Table 3).

Of the total participants, 19 (23%) of them didn't practice stay at home to prevent COVID-19. This figure is also true for social distancing. About 23 (28%) of the clients kept their social distances with less than two meters. Nearly 16 (20%) of the study subjects involved in social gatherings after the outbreak of COVID-19 (Fig 1).

## Discussion

The current research aims to explore the level of understanding and strategies to prevent the pandemic after commencement in Ethiopia. For people to survive in the period of pandemic, sufficient knowledge of the virus that caused the pandemic is mandatory.

### Knowledge of clients about COVID-19

Our study revealed the majority of respondents had good knowledge about the symptoms, way of spread, and its prevention strategies of the virus. In the current study, about 69.12% of

**Table 2. Knowledge of clients towards COVID-19 in Dessie town public health facilities, Ethiopia, 2020 (n = 81).**

| Variables | Category | Frequency | Percent (%) | Score |
|---|---|---|---|---|
| **Knowledge of Cause** | Virus | 47 | 69.12* | 1 |
| | Other than virus | 21 | 30.28 | 0 |
| **Knowledge of treatment and high risk** | | | | |
| Presence of treatment/vaccine | Yes | 9 | 11.11 | 0 |
| | No | 72 | 88.89* | 1 |
| Are old age and co-morbidity risk conditions for COVID-19 | Yes | 63 | 77.78* | 1 |
| | No | 18 | 22.22 | 0 |
| **Knowledge of transmissions and infectiousness** | | | | |
| Contagious | Yes | 79 | 97.53* | 1 |
| | No | 2 | 2.47 | 0 |
| Transmitted through Sneezing | Yes | 76 | 93.83* | 1 |
| | No | 5 | 6.17 | 0 |
| Transmitted through coughing | Yes | 78 | 96.30* | 1 |
| | No | 3 | 3.70 | 0 |
| Transmitted through contact | Yes | 78 | 96.30* | 1 |
| | No | 3 | 3.70 | 0 |
| Transmitted through shaking | Yes | 65 | 80.25* | 1 |
| | No | 16 | 19.75 | 0 |
| Transmitted through insect bite | Yes | 12 | 14.81 | 0 |
| | No | 69 | 85.19* | 1 |
| Transmitted through blood | Yes | 13 | 16.05* | 1 |
| | No | 68 | 83.95 | 0 |
| Knowledge of transmissions score | | | | **4.68±0.63** |
| **Knowledge of symptoms** | | | | |
| Is sneezing COVID-19 symptoms | Yes | 72 | 88.89 | 0 |
| | No | 9 | 11.11* | 1 |
| Is fever COVID-19 symptoms | Yes | 78 | 96.30* | 1 |
| | No | 3 | 3.70 | 0 |
| Is difficulty of breathing COVID-19 symptoms | Yes | 71 | 87.65* | 1 |
| | No | 10 | 12.35 | 0 |
| Is cough COVID-19 symptoms | Yes | 78 | 96.30* | 1 |
| | No | 4 | 3.70 | 0 |
| Knowledge of symptoms Score | | | | **3.69±0.77** |
| **Knowledge about ways of prevention** | | | | |
| Is isolation prevent from COVID-19 | Yes | 57 | 70.37* | 1 |
| | No | 24 | 29.63 | 0 |
| Is physical distancing prevent from COVID-19 | Yes | 76 | 93.83* | 1 |
| | No | 5 | 6.17 | 0 |
| Hand hygiene prevent from COVID-19 | Yes | 76 | 93.83* | 1 |
| | No | 5 | 6.17 | 0 |
| Is stay at home prevent from COVID-19 | Yes | 71 | 87.65* | 1 |
| | No | 10 | 12.35 | 0 |
| Is face mask prevent from COVID-19 | Yes | 64 | 79.01* | 1 |
| | No | 17 | 20.99 | 0 |
| Does avoid social gathering prevent COVID-19 | Yes | 62 | 76.54% | 1* |
| | No | 19 | 23.46% | 0 |
| Knowledge about ways of prevention score | | | | **5.91±1.43** |
| Total score | | | | 18 |

(*Continued*)

**Table 2.** (Continued)

| Variables | Category | Frequency | Percent (%) | Score |
|---|---|---|---|---|
| Min-max | | | | 6–16 |
| Mean ± SD | | | | **14.38±1.85** |

*correct answer.

**Table 3.** Practices of the clients towards COVID-19 in Dessie town public health facilities, Ethiopia, 2020 (n = 81).

| Variables | Response | Participant (%) | Knowledge | | Chi-square | P |
|---|---|---|---|---|---|---|
| | | | Poor(<median) | Good(≥median) | | |
| Do you stay at home | Yes | 62 (76.54) | 10 | 46 | - | - |
| | No | 19(23.46) | 25 | 0 | | |
| Are you using mask | Yes | 64(79.01) | 14 | 3 | 13.43 | 0.001 |
| | No | 17(20.99) | 21 | 43 | | |
| Avoid social gathering | Yes | 65(19.75) | 21 | 5 | 9.39 | 0.002 |
| | No | 16 (80.25) | 14 | 41 | | |
| Practice of social distancing | <2 meters | 23(28.40) | 10 | 25 | 0.001 | 0.97 |
| | ≥2 meters | 58(71.60) | 13 | 33 | | |
| Regular hand washing | Yes | 79 (97.53) | 5 | 3 | 1.35 | 0.25 |
| | No | 2 (2.47) | 30 | 43 | | |
| Isolation | Yes | 57(70.37) | 21 | 3 | 27.26 | 0.001 |
| | No | 24(29.63) | 14 | 43 | | |
| Overall preventive practice | Poor(< median) | 28(34.57) | 23 | 5 | **26.43** | **0.001** |
| | Good(> = median) | 53(65.43) | 12 | 41 | | |

the participants were reported that the disease is caused by virus. Similarly, the majority (88.89%) of the participants were reported that the disease has no definitive cure. Almost all respondents about, (97.53%), of them reported that the disease is highly contagious. About 77.78% of the respondents have recognized the disease become more severe with chronic comorbidity diseases and elderly people.

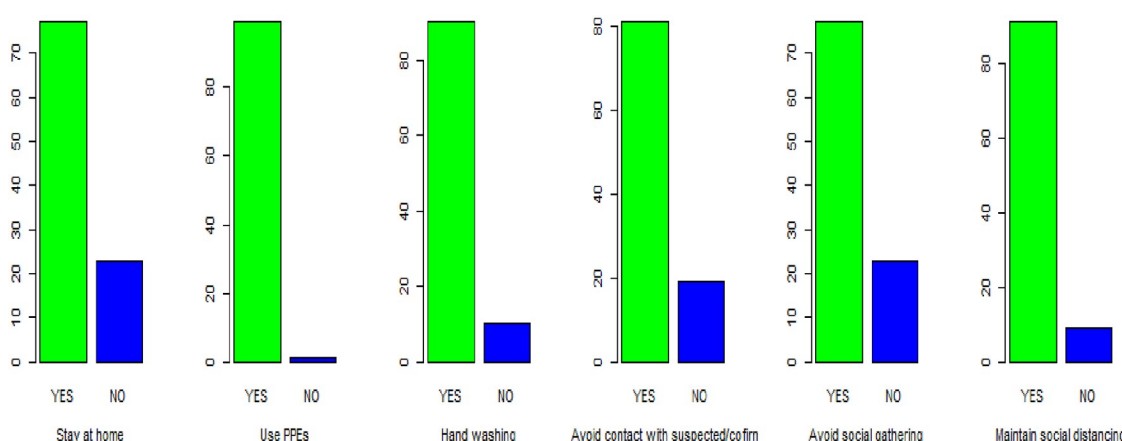

**Fig 1. Practices of clients towards COVID-19 prevention measures in Dessie town public health facilities, Ethiopia, 2020 (n = 81).**

Despite this result parallel with study findings [12, 13], reports also show mortality from COVID-19 include all age groups [14]. This implies sufficiently addressing young people in pandemic messaging is required. However, almost all (93.83%) participants have misconceptions regarding symptoms like sneezing as symptom of COVID-19 which shown in common cold. In many similar epidemics such misconception leads to stigma [15, 16]. This stigma could lead to severe difficulties for controlling a COVID-19 pandemic through hiding people complaint to avoid discrimination and prevent seeking health care immediately. The WHO endorses that any message and case management struggles should concurrently address the stigma associated with COVID-19 by providing do's and don'ts [17]. This implies that all stakeholders are anticipated to assimilate stigma reduction interventions across all COVID-19 fighting activities.

Client's knowledge score has no significant difference among sex, education status, and monthly income. This finding is concurrent with studies done in different parts of Africa [18]. This might be due to global, national and local stakeholders and charity groups effort on awareness creation activity as well as ability of the government to make COVID-19 a community concern of the era. However, COVID-19 transmission knowledge scores significantly higher among urban residents. This finding is synchronized with study findings [18, 19]. This might be related to the gap to address wide area of rural residences in their scattered habitat and poor access of media as well.

Even though, the Ministry of health (MOH) of Ethiopia disseminates COVID-19 related information through social Medias, the main source of information for current study participants was broadcast platform for more than 87.65% of clients at the expense of more modern platforms. These platforms provide reliable information than those an easily accessible social media sources. With caution for spread of fabricated information in addition broadcast social Medias are alternative sources of fast ad easily accessible source of information [20].

## Strategies for prevention of the pandemic virus

The mean score of strategies towards prevention of the pandemic was 4.75 with standard deviation (±1.28 from 6 components). The mean score of COVID-19 preventive practice varies across different COVID-19 transmission knowledge (P-value = 0.001) and COVID-19 symptom knowledge (P-value = 0.0012).

This study showed an overall preventive practice was 64.20% implementing six or more components as required. However, each component practices as high as 98.77% use personal protective equipment and hand washing (97.53%) regularly. On the other hand staying at home practice and avoid social gathering (76.54%) maintained minimum two meters distance was (71.60%). Possibly, in developing countries, numerous causes can be mentioned as to why people cannot easily avoid social gathering and staying at home. For instance, a high rate of overcrowded living conditions, frequent social and religious ceremonies, absence of reserve food for extended time [21].

Individuals more knowledgeable about way of transmission and symptoms of COVID-19 had significantly higher COVID-19 preventive practice. This result is consistent with previous reports from China [21] and Ethiopia [12]. This result clearly indicates the knowledge of the symptoms and way of transmission also associated with acquiring the knowledge of the contagiousness of the disease and way of preventions as well. This study clearly indicated the significance of enhancing COVID-19 knowledge through different media that would have a role of enhancing the preventive strategies of the virus.

The strength of this study lies in its novelty in the study area and conducted in the early stage of the COVID-19 pandemic in Ethiopia. This also helps to ignite the stakeholder's

activity to halt the expansion of the pandemic. However, the finding had limitations like, small sample size, institutional based study, and over representation of urban residents will compromise its representativeness. So, our findings can only be generalized to urban populations of a relatively have access to timely information.

Due to inadequate access of information in rural residents and individuals in the community, there will be vulnerable populations for COVID-19 pandemic as a result of having poor knowledge about way of transmission and unfortunate strategies towards COVID-19 prevention. Therefore, knowledge and practice towards COVID-19 among people in rural dwellers deserves special research consideration in today's Ethiopia.

## Conclusion

As conclusion, our findings suggest that good COVID-19 knowledge about way of transmission and symptoms was associated with proper preventive practices about the pandemic, signifying that health education packages directed at improving COVID-19 knowledge is useful for combating COVID-19 and continuing safe practices. Attributable to the limitation in representativeness of the sample, further studies are necessary using mixed study design with large sample size among high risk communities.

## Supporting information

**S1 Dataset.**
(DTA)

## Acknowledgments

We would like to thank the data collectors, supervisors and health facility managers for their cooperation during data collection for the accomplishment of this research work.

## Author Contributions

**Conceptualization:** Reta Dewau.

**Data curation:** Reta Dewau, Tefera Chane Mekonnen, Sisay Eshete Tadesse, Amare Muche, Getahun Gebre Bogale, Erkihun Tadesse Amsalu.

**Formal analysis:** Reta Dewau, Tefera Chane Mekonnen, Sisay Eshete Tadesse, Amare Muche.

**Investigation:** Reta Dewau.

**Methodology:** Reta Dewau, Tefera Chane Mekonnen, Sisay Eshete Tadesse, Amare Muche, Erkihun Tadesse Amsalu.

**Software:** Sisay Eshete Tadesse, Erkihun Tadesse Amsalu.

**Supervision:** Amare Muche.

**Validation:** Reta Dewau, Getahun Gebre Bogale.

**Visualization:** Erkihun Tadesse Amsalu.

**Writing – original draft:** Reta Dewau, Tefera Chane Mekonnen, Sisay Eshete Tadesse, Amare Muche, Getahun Gebre Bogale, Erkihun Tadesse Amsalu.

**Writing – review & editing:** Reta Dewau, Tefera Chane Mekonnen, Sisay Eshete Tadesse, Amare Muche, Getahun Gebre Bogale, Erkihun Tadesse Amsalu.

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
