## [Decision Letter · Decision Letter 0]

9 Dec 2020

PONE-D-20-23137

Knowledge and Practices of Clients on preventive measures of COVID-19 Pandemic among Governmental health facilities in South Wollo, Ethiopia: A Facility-based Cross-Sectional study

PLOS ONE

Dear Dr. Tadesse,

Thank you for submitting your manuscript to PLOS ONE. After careful consideration, we feel that it has merit but does not fully meet PLOS ONE’s publication criteria as it currently stands. Therefore, we invite you to submit a revised version of the manuscript that addresses the points raised during the review process.

I request you to review the comments and suggestions given by the reviewers and come up with a revised version incorporating those line by line.

We look forward to receiving your revised manuscript.

Kind regards,

Animesh Biswas, PhD, MPH, MSc, MBBS

Academic Editor

PLOS ONE

Journal Requirements:

2.We suggest you thoroughly copyedit your manuscript for language usage, spelling, and grammar. If you do not know anyone who can help you do this, you may wish to consider employing a professional scientific editing service.  

Reviewers' comments:

Reviewer's Responses to Questions

**Comments to the Author**

1. Is the manuscript technically sound, and do the data support the conclusions?

Reviewer #1: Yes

Reviewer #2: Yes

2. Has the statistical analysis been performed appropriately and rigorously? 

Reviewer #1: Yes

Reviewer #2: I Don't Know

3. Have the authors made all data underlying the findings in their manuscript fully available?

Reviewer #1: Yes

Reviewer #2: Yes

4. Is the manuscript presented in an intelligible fashion and written in standard English?

Reviewer #1: Yes

Reviewer #2: Yes

5. Review Comments to the Author

Reviewer #1: The sampling method not clear in relation to facility selection and randomization. Number of sample is relatively low to be representing. Use of old census for the catchment area for sample calculation without estimation of expected growth affects the result generalization. selection biase due to facility base sampling does not support Community base result.

Reviewer #2: Currently Covid-19 related articles have become one of the most popular topics. Design of the study - cross-sectional. Statistical part written good, used statistical tests described, approved with the tables and graphs. There is some concern about sample size (82) and duration of the study. Is it enough for generalizing the result with that sample size, and why the duration only 9 days??? For that reason the question: Has the statistical analysis been performed appropriately and rigorously? - answered Don't know. The article itself well written, English in submitted article is clear, understandable, article style used.

6. PLOS authors have the option to publish the peer review history of their article (what does this mean?). If published, this will include your full peer review and any attached files.

Reviewer #1: No

Reviewer #2: No

---

## [Author Response · Author response to Decision Letter 0]

18 Dec 2020

Response to reviewer Date: 16 December/2020

PONE-D-20-23137

Knowledge and Practices of Clients on preventive measures of COVID-19 Pandemic among Governmental health facilities in South Wollo, Ethiopia: A Facility-based Cross-Sectional study

Erkihun Tadesse

PLOS ONE

Dear all,

We would like to thanks for these constructive, building and improvable comments on this manuscript that would improve substance and content of the manuscript. We considered each comments and clarification questions of editors and reviewers on the manuscript thoroughly. Our point-by-point responses for each comment and questions are described in detailed on the following pages. Further, the details of changes were shown by highlighting in the revised manuscript file attached.

Editor comments 

1. Please ensure that your manuscript meets PLOS ONE's style requirements, including those for file naming. The PLOS ONE style templates can be found:

https://journals.plos.org/plosone/s/file?id=ba62/PLOSOne_formatting_sample_title_authors affiliations.pdf

Response: Thank you editor for the supportive comment. Our manuscript meets PLOS ONE's style requirements.

2. We suggest you thoroughly copyedit your manuscript for language usage, spelling, and grammar. 

Response: Thank you very much editor for the supportive suggestions to improve our work for readability. We amended the language, spelling and grammar in the revised manuscript file shown by highlighting. 

Response: Thank you very much editor for the supportive comments to improve our work. We take the ethics statement to the methods section and deleted it from the declaration section in the revised manuscript file shown by highlighting. 

Response: Thank you very much editor for the supportive comment and we included caption for figure 1 and figure 2 in the manuscript file as shown by highlighting. 

Reviewer 1 comments 

1. The sampling methods not clear in relation to facility selection and randomization. Number of sample is relatively low to be representing. Use of old census for the catchment area for sample calculation without estimation of expected growth affects the result generalization. Selection bias due to facility base sampling does not support community base result. 

Response: Thank you reviewer for the constructive and supportive comments to improve our work for readability. We clarified the sampling methods in relation to sample size determination, facility selection and randomization in the revised manuscript file from line number 115-129.

-“We calculated the sample size using score test for one sample proportion by STATA version 14 with the assumption that 50% of clients had good knowledge that scored above the mean, 5% level of significance (α), power 80% (β), and at least 16% of client’s knowledge improvement is expected after six months of the pandemic duration (December to May). Thus, the total calculated sample size was 83 clients.”

-“We selected Dessie town purposively from south wollo zone since it is hotspot for the spread of the virus as speculated by the Ethiopian Public Health Research Institute and selected by the federal ministry of health of Ethiopia as the only treatment and screening center for COVID-19 during the early phase of the pandemic in the country. We selected 3 health facilities namely Dessie Referral hospital, Segno Gebeya health center, and Buanbuawuha health centers randomly using lottery method. Then, we proportionally allocated the sample size to each health facility based on the patient flow of the facilities. Finally, we selected 43 participants from Dessie referral hospital, 20 participants from Segnogebeya health center, and 20 participants from Buanbuawuha health center consecutively.” 

 As a limitation of this study we considered small sample size in which the finding from the study may not be representative/ not enough for generalization. Since we conducted this study during the early phase of the pandemic in the country, Ethiopia where there is no evidence (study) about the level of knowledge and preventive practice of the COVID-19 pandemic in the country including our study area. Thus, we conducted this study within 9 days of data using 81 participants attending the selected health facilities to provide urgent evidence (feedback) to the concerned bodies to halt further progression of the pandemic by designing appropriate prevention and control strategies.

Reviewer 2 comments

-Statistical part written good, used statistical tests described, approved with the tables and graphs. There is some concern about sample size (81) and duration of the study. Is it enough for generalizing the result with that sample size, and why the duration only 9 days ???

For that reason the question: Has the statistical analysis been performed appropriately and rigorously? - answered don’t know. The article itself well written, English in submitted article is clear, understandable, article style used. 

Response: Thank you reviewer for the constructive and supportive comments to improve our work for readability. As a limitation of this study we considered small sample size in which the finding from the study may not be representative/ not enough for generalization and described in the discussion section of the manuscript file. 

We conducted this study during the early phase of the pandemic in the country, Ethiopia where there is no evidence (study) about the level of knowledge and preventive practice of the COVID-19 pandemic in the country including our study area. In spite of this limitation, we conducted this study within 9 days of data using 81 participants attending the selected health facilities to provide urgent evidence (feedback) to the concerned bodies to halt further progression of the pandemic by designing appropriate prevention and control strategies.

---

## [Editor Report · Decision Letter 1]

11 Feb 2021

Knowledge and Practices of Clients on preventive measures of COVID-19 Pandemic among Governmental health facilities in South Wollo, Ethiopia: A Facility-based Cross-Sectional study

PONE-D-20-23137R1

Dear Dr. Tadesse Amsalu,

We’re pleased to inform you that your manuscript has been judged scientifically suitable for publication and will be formally accepted for publication once it meets all outstanding technical requirements.

Kind regards,

Animesh Biswas, PhD, MPH, MSc, MBBS

Academic Editor

PLOS ONE
---

## [Editor Report · Acceptance letter]

12 Feb 2021

PONE-D-20-23137R1 

Knowledge and Practice of clients on preventive measures of COVID-19 Pandemic among Governmental health facilities in South Wollo, Ethiopia: A Facility-based Cross-Sectional study 

Dear Dr. Tadesse Amsalu:

I'm pleased to inform you that your manuscript has been deemed suitable for publication in PLOS ONE. Congratulations! Your manuscript is now with our production department. 

Kind regards, 

on behalf of

Dr. Animesh Biswas 

Academic Editor

PLOS ONE